# Effects of the Head Start Program on Socioecological Obesogenic Factors in American Children

**DOI:** 10.3390/ijerph18094779

**Published:** 2021-04-29

**Authors:** Taeeung Kim, Minju Kim, Chang-Yong Jang, Nam-Gyeong Gim

**Affiliations:** 1Department of Epidemiology, University of California, Irvine, CA 92697, USA; ktang7711@gmail.com; 2Department of Dance, Hanyang University, Seoul 04763, Korea; amy1206@daum.net; 3Department of Physical Education, Andong National University, Andong 36729, Korea; 4Department of Administration, Yuk-buk Elementary School, Yongin 17061, Korea

**Keywords:** Head Start, children, quality of life, socioecological obesogenic factors

## Abstract

Head Start is a nationwide developmental program for low-income families. This study aimed to investigate the association between the Head Start program and children’s BMI status, as well as their quality of life with respect to socioecological obesogenic factors. This cross-sectional study employed the Early Childhood Longitudinal Study-Kindergarten cohort (ECLS-K) in which the data were collected in 2007 and analyzed in 2019. Propensity-score matching analysis was performed to examine the association between the Head Start program and children’s BMI status, as well as the quality of life, controlling for socioecological obesogenic factors. A total of 3753 children (representing 1,284,209 at the population level) were recruited in this study (mean age: 13.69 years; girls: 49.42%). In the final matched model, the program did not have a statistically significant effect on children’s obesity. Fewer African American children participated in school-sponsored activities, perceived themselves as overweight, lived in a household with fewer family members, had less strict TV regulations, and were more likely to be overweight than their counterparts. Outcomes suggest that multiple dimensions of sociological obesogenic factors including individual, parental, familial, and community support factors affect the weight of children from low-income families and should be considered when establishing behavioral and policy interventions to thwart the childhood obesity epidemic.

## 1. Introduction

To prevent and defer obesity during childhood, and later in the adult life, it is necessary to understand the complexities and opacities of obesity and related behaviors in order to develop more effective and efficient anti-obesity interventions. In the last two decades, given the scope of the issue, many city- and state-level obesity-prevention efforts and campaigns have been proposed and enacted, encompassing the most successful legislation and laws on both childhood obesity and obesity in adults [1,2,3,4]. 

More than one-third of the adults in the United States (U.S.), representing over 72 million people, are overweight and/or obese [5,6] and children show a similar trend. The prevalence of obesity and associated trends among US children and their negative mental/psychological [7] and physical [8] consequences, both during childhood and later in their adult life [9], have been well established and documented [10]. 

Factors reported to contribute to the increase in childhood obesity include lower physical activity [11,12], higher caloric intake [12,13], sedentary lifestyle [14], and a combination of genetic, family, community, and socioeconomic factors (e.g., level of income, education, gender, and living conditions) [15]. Such poor lifestyle patterns are substantially attributed to families [16], schools [3], communities [2], and environments [17]. Schools and communities can play an important role in preventing and minimizing childhood obesity by establishing healthy and positive environments with policies and initiatives that provide opportunities for school-aged children to learn and practice healthy eating and become more physically active [18]. 

Head Start is a nationwide developmental program for low-income families and is one of the most prominent policies and intervention methods to improve education, well-being, and quality of life of children [19]. It provides comprehensive well-being measures such as education, nutrition and health information, and parental involvement services for children and their low-income families through agencies in the local community. The long-term effects of the Head Start program on children from low-income families [20] have not yet been confirmed. However, it can help in bridging the knowledge gap between the federal interventional regulations and their implementation in terms of a child’s social well-being and risk for obesity [21]. The program also showed favorable associations among nutrition services, education, and children’s weight reduction, helping to build the right eating attitudes through positive family-involved meals and practices [22]. 

However, there is still a paucity of data on the association between the Head Start program and children’s BMI status, quality of life, and other socioecological obesogenic factors, such as participation in structured physical activities [23,24,25] or eating fewer family meals [26], in a large nonclinical sample of children. Since little is known regarding the effectiveness of well-being programs associated with the prevention of childhood obesity, such a study would contribute to the literature and the field.

## 2. Materials and Methods

### 2.1. Data Source

The data for this study were obtained from the Early Childhood Longitudinal Study-Kindergarten cohort (ECLS-K), which followed children from kindergarten (the 1998 to 1999 school year) to fifth grade (through the 2007 school year) (Table 1). The 1998 to 1999 kindergarten class cohort is a sample of children from kindergarten through eighth grade, which is a nationally representative sample of kindergartners, parents, teachers, and schools from across the US. General eligibility requirements for the Head Start and Early Head Start programs are children aged zero to five years with families below the federal poverty level (FPL) [27,28]. Therefore, respondents who enrolled in Head Start in Wave 1 (1998 to 1999) and Wave 2 (1999 to 2000) and children who were not enrolled in Head Start were analyzed. Participants who dropped out of the program at any time were not considered for analysis in this study.

### 2.2. Conceptual Frameworks

As shown in Figure 1, a modified socioecological conceptual model [29] was employed as a theoretical framework for this study to better understand the complexity of early childhood obesity. The development of the assessment framework consists of five main parts: (1) a child’s individual factors such as age, gender, race/ethnicity, health, and self-care skills; (2) parenting function, which includes education, employment status, parental personalities and characteristics, and income level; (3) family function, which includes family structure, family size, level of poverty, family regulation, and security; (4) school factors such as type of school, ratio of minority students, and school lunch programs; and (5) environmental factors such as geographic location and urbanity. 

### 2.3. Dependent Variable

The main binary variable outcome was being overweight (e.g., ≥85th BMI percentile) versus normal weight (e.g., 5th ≤ BMI percentile < 85th) for age and gender, respectively [30]. Children’s BMI was calculated as weight in kilograms divided by square of height in meters [31]. The BMI percentile was used to determine the severity of childhood obesity based on the age and gender growth chart [32].

### 2.4. Independent Variables

The five major socioecological factors related to early childhood obesity are as follows: individual factors, parenting capacity, family function, school factors, and environmental factors. First, young children’s individual obesity-related variables consist of age, gender, race/ethnicity, computer usage, and the number of hours of watching TV after dinner. Participants ranged in age from 2 to 15 years old. Participants’ race/ethnicity was categorized into four groups: Hispanic, non-Hispanic white, non-Hispanic Black, and ‘Other’. Computer usage was assessed as the number of times per week measured on a 4-point Likert scale from 1 (never) to 4 (daily). Watching TV after dinner was rated as the average amount of time they watch TV or videos from 0 to 7 h each day at home. Participation in a school-sponsored activity was assessed based on the number of hours of children’s participation each school year. The binary variable measures whether children participated in a sports activity. A 4-point Likert scale was used to see whether the intent to control their weight was influenced by Head Start, ranging from 1 (no intention) to 4 (full intention to control weight). Children’s perceived obesity status was assessed using a 4-point Likert scale to determine the extent to which they described themselves as overweight/slightly overweight or obese, from 1 (no obesity) to 4 (fully perceived to be overweight). The level of sedentary lifestyle was evaluated using a composite score for comprehensive sedentary behaviors: how many hours per week they (a) typically watch TV, videos, or DVDs, (b) play computer or video games (e.g., Nintendo, PlayStation, or Xbox), and (c) are on the internet (e.g., e-mailing/instant messaging with friends, or surfing the Web)

The parental variables related to childhood obesity in this study consisted of four factors. The education level of householders was classified into three groups, from high school or lower to high school or higher. Parents’ employment was measured as a composite score for parents’ weekly working hours (1 = no labor to 4 = 35 h per week). Parents’ health was calculated using their mental and physical health status. A composite score was obtained for comprehensive parent-child relationships, such as talking with the child each day about school, helping with schoolwork, or advising children on important decisions. 

Family functions consisted of seven variables. The family structure associated with the parent of the child was restructured as a binary variable (e.g., two parent families versus another family structure). The health of a child with two parents may differ from the health of another child in other family structure [16,33]. The 7-point Likert scale for the primary care type was reconstructed using ‘0’, which indicates parental care, and ‘1’, which indicates parental care, as binomial variables, and children who received parental care may have different health performance compared to children taken care of by others. [34,35,36,37]. The number of families under the age of 18 was measured in the range of 1–11. Nine classification categories for primary caregivers of children living in the home were measured and reconstructed into binary variables, including biological parents versus others. This is because children being cared for by biological parents may have different health outcomes than their counterparts [37,38,39]. Participants’ income was measured from less than USD 5000 to more than USD 200,001 on a 12-point Likert scale. In addition, income levels were classified as quintile indicators to secure unbiased measurements of income levels. A binary variable for family rules was used to measure TV watching restrictions for children. Food security in the home was assessed on a 4-point Likert scale, from food insecurity with hunger to food security. Finally, the SES of the family was measured on a quintile basis for socioeconomic status.

One of the main environmental obesity factors in childhood obesity is the role of schools where children nest [40]. School environment factors are the percentage of minority students and the percentage of students who can get lunch at free or discounted prices. The percentage of ethnic minority students was rated on a 5-point Likert scale from 1 (less than 10%) to 5 (more than 75%). To measure the percentage of students eligible for reduced-price lunch service, a 5-point Likert scale was used from 1 (less than 1%) to 5 (more than 25%). Finally, the characteristics of children’s geographic residence were classified into three urbanities: large cities, small and medium-sized cities, and small towns and rural areas. 

### 2.5. Satistical Analyses

Pearson’s χ2 tests and t-tests with weighted counts and column percentages were performed for descriptive statistics. A propensity score matching (PSM) analysis [41] was performed to minimize selection bias since Head Start enrollees are from relatively low-income families. Logistic regression [42] was used to examine the association between the Head Start program and children’s BMI status and quality of life in terms of socioecological obesogenic factors. All statistical analyses and the logistic regression analysis were conducted on gender, race/ethnicity, SES, geography, and/or other relevant characteristics using the STATA version 15.1 and SAS version 9.4.

## 3. Results

This study examined the effects of the Head Start program on children’s BMI status and their quality of life in terms of socioecological obesogenic factors. Contrary to our hypothesis, the Head Start program did not have a statistically significant effect on children’s obesity. The unweighted (e.g., number of participants) and weighted (e.g., mean, SD, and percentage) descriptive statistics of the study population are shown in Table 2. A total of 3753 children (representing 1,284,209 at the population level) were recruited for this study. The mean age was 13.69 years, and while 49.42% of the participants were girls, only 448 (15.11%) of them participated in the Head Start program. As shown in Table 2, the participants belonged to diverse races, including non-Hispanic Whites (60.04%), Hispanics (18.75%), non-Hispanic Blacks (14.41%), and Others (6.75%).

### 3.1. Imbalance Test of PSM for the Pair-Matching Procedure

As seen in Table 3; the pair-matching procedure matched 351 participants from the Head Start Program to 351 nonparticipants. The matches in this study sample did not differ significantly on the observed covariates.

### 3.2. Imbalance Test of PSM with a Comparison between Matched and Unmatched Covariates

Seven independent variables associated with children’s overweight status had significantly different odds ratios (ORs) in their binary BMI levels (i.e., overweight (≥85th percentile) vs. healthy weight (<85th percentile)) (see Table 4). The Head Start program did not have a statistically significant effect on children’s obesity in this final matched model. More detailed study results are discussed below.

Regarding the child’s individual obesogenic factors, non-Hispanic American Blacks were at a higher risk of being overweight compared to non-Hispanic whites (OR = 5.23, 95% CI = 2.04–13.37). A one-unit increase in the number of hours of participation in school-sponsored activities during the school year resulted in a lower risk of being overweight (OR = 1.07, 95% CI = 1.01–1.15). Conversely, children who participated in sports activities were less likely to be overweight than those who did not (OR = 0.32, 95% CI = 0.17–0.61). A one-unit increase in negative perceived weight increased a child’s odds of being overweight (OR = 6.86, 95% CI = 4.22–11.16). In terms of children’s willingness to change their weight, the risk of obesity was significantly lower among children who wanted to increase weight, children who wanted to keep the same weight, and those who did not try anything to change their weight (OR = 0.04, 95% CI = 0.01–0.15; OR = 0.08, 95% CI = 0.04–0.16; and OR = 0.15, 95% CI = 0.08–0.30, respectively). Lastly, with respect to the family function related to children’s BMI status, an increase in one family member under the age of 18 resulted in a reduced risk of being overweight (OR = 0.78, 95% CI = 0.63–0.97). Children with TV restrictions in their households had a lower risk of being overweight than those who did not (OR = 0.40, 95% CI = 0.20–0.80).

## 4. Discussion

The purpose of this study was to determine if the Head Start program affects children’s BMI status with respect to socioecological obesogenic factors through a PSM analysis. Table 4 shows how the Head Start program affects children’s BMI status before and after PSM, controlling for sociodemographic factors. Both the models indicated that the program did not significantly contribute to children’s weight. On the other hand, though not statistically significant, it was associated with a decrease in children’s weight.

Although the program provides early childhood education services for approximately one million low-income children in more than 2000 local branches nationwide with the support of federal funds, one-third of the children participating in the program are obese or overweight [43]. For purposes of policy enforcement and the implementation of childhood anti-obesity policies, directors of the Head Start program have addressed three challenges: lack of time, funding issues, and lack of knowledge. Some low-income parents believe that early childhood obesity is healthy [44,45], making it difficult to implement childhood anti-obesity policies. 

Despite the research results discussed above and doubts about the efficiency of the Head Start program, important reasons exist for continuing to develop the program for the physical and cognitive health of children. First, nutrition education and other services of the program can contribute to minimizing and preventing childhood obesity as a large anti-obesity campaign for children from low-income families [21]. The Head Start program was associated with children’s weight loss through nutrition services and education, improvements in the healthy eating habits of children through mealtimes, and an environment in which family members can participate [46]. Full-day Head Start care services may significantly help in reducing or delaying the rapidly increasing rates of overweight and/or obese children [47]. 

A significant association between Head Start and disparities in childhood obesity also was shown in terms of SES, i.e., a high proportion of overweight and obese children from low-income Hispanic families that speak English [48]. In addition, there is a much more significant decrease in the childhood obesity rate in Head Start programs located in city centers. A nationwide survey [45] evaluated the associations between enrollees in the Head Start program and their BMI status, since the program encouraged and promoted healthier eating habits (e.g., only nonfat or 1% fat milk, some fruit rather than fruit juice each day, some vegetables rather than fried potatoes, and offering special events with healthy foods or nonfood treats) and being more physically active (e.g., adult-led or adult-guided structured gross motor activities and/or unstructured gross motor activities for at least 30 min each day). These associations meet the federal requirements of daily needs compared to other children’s well-being and welfare programs.

The effects of direct weight loss (e.g., BMI) have been shown in children enrolled in a Michigan Head Start program, especially among three- to five-year-old girls from racial/ethnic minorities [49]. The participants were relatively less obese, less overweight, and less underweight at follow-up [50]. This may be an alternative anti-obesity interventional system for preschoolers with an unhealthy weight that can be used to fulfill the role of an effective prekindergarten, obesity-suppression program compared to other primary care systems.

Educating children and parents about healthier eating behaviors and nutrition services led to the suppression and reduction of obesity in Black children [21]. 

In addition, one interesting study analyzing the long-term effects of the Head Start program was based on cost-effectiveness [51]. Poor short-term effects of the Head Start programs on the physical and mental development of children have been reported in numerous studies [22,45,48,52,53,54], but the long-term effects have received little academic attention. Although there are a few short-term impacts of the program, it has proven to be a cost-effective child development policy from a long-term perspective in early participant cohorts [20,51].

Given that the extent of the effectiveness of children’s well-being programs associated with childhood obesity is quite controversial, the present study contributes to the literature and to the field. Some debate exists about the physical and cognitive effects of the Head Start program. While short-term effects may be insignificant or show a negligible impact on outcomes, it is arguable that the long-term effects on children’s well-being will positively affect the welfare of children from low-income families. A more systematic and diverse program for physical and cognitive development, consistent nutrition services, education, and the active participation of the government is needed to maximize the effects of the Head Start program on child welfare. 

This study should be interpreted in light of the following limitations. First, due to the data characteristics, interpretation of a causal relationship between childhood obesity and socioecological obesogenic factors is limited. Second, this study utilized self-reported measurements to assess all study variables from respondents’ one-day recollection, which may be inaccurate due to recall bias, respondent bias, or interview bias. Third, the small sample size (e.g., non-white groups) cautions against generalizing the research findings to those living in more diversely populated areas. A larger sample size with more diverse participants is likely to yield better and more reliable results. Finally, unobserved confounding socioecological factors related to childhood obesity might have been missed, decreasing the matching efficiency of the PS. However, these limitations do not outweigh the contributions of this study. Future studies should also examine if multilevel effects can contribute to children’s BMI status.

## 5. Conclusions

This study revealed how different sociological factors influence children’s BMI status in a large sample of US children. Individual, familial, and environmental factors were found to affect children’s BMI status. Additionally, achieving success through the childhood anti-obesity policies in the Head Start program requires increasing federally funded staff training and high-tech supplementation, as well as offering healthier and more nutritionally balanced snacks to children. 

Multifaceted socioecological obesogenic factors affect children’s weight. Policymakers and practitioners should consider the children’s racial/ethnic characteristics in anti-obesity interventions, in order to minimize and/or eliminate disparities in health services and well-being programs for minorities. An understanding of the role of families, schools, and communities in terms of how they contribute to children’s obesity status, as well as the role of socioecological factors, are important if we wish to prevent and minimize the pandemic of obesity when children are flourishing. Therefore, health educators, professionals, policymakers, and stakeholders must consider a multidimensional approach when it comes to committing to and implementing intervention programs or policies aimed to improve the quality of life of children.

## Figures and Tables

**Figure 1 ijerph-18-04779-f001:**
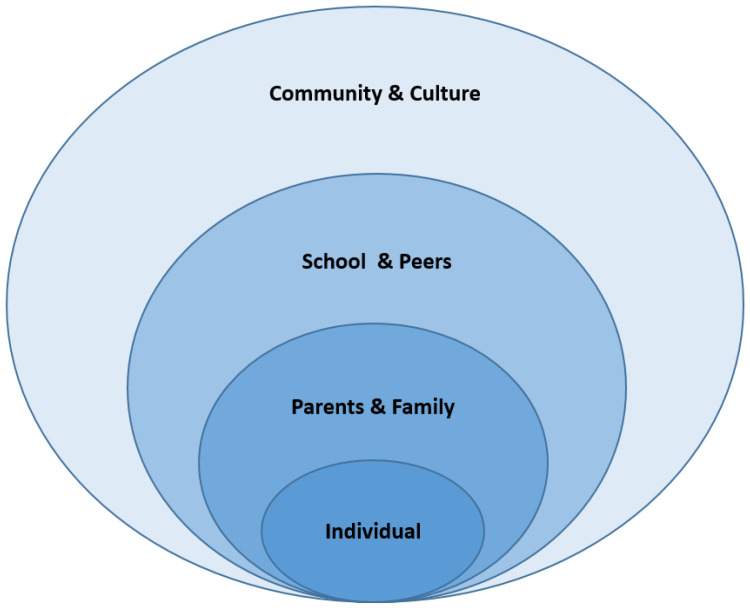
Modified socioecological systems theory (Adapted from Bronfenbrenner [29]).

**Table 1 ijerph-18-04779-t001:** ECLS-K waves of data collection.

Data Collection	Date of Collection	Sample
Fall—kindergarten	Fall 1998	Full sample
Spring—kindergarten	Spring 1999	Full sample
Fall—first grade	Fall 1999	30% subsample ^1^
Spring—first grade	Spring 2000	Full sample plus freshening
Spring—third grade	Spring 2002	Full sample
Spring—fifth grade	Spring 2004	Full sample
Spring—eighth grade	Spring 2007	Full sample

^1^ Fall data collection consisted of a 30% sample of schools containing approximately 27% of the base-year children.

**Table 2 ijerph-18-04779-t002:** Weighted descriptive statistics of study sample with overweight status.

			% (*n*), Mean (SD)			
Variables			Overweight	Normal Weight	Overall	*p*-Value
Dependent variables	Overweight	Binary variable (1 = overweight, 0 = normal)	33.20% (1252)	66.80% (2501)	100% (3753)	
Treatment	Head Start	Binary variable (1 = yes, 0 = no)	17.80% (191)	13.75% (257)	15.11% (448)	0.05
Covariates	1. Individual					
		Age	13.67 (0.54)	13.70 (0.51)	13.69 (0.52)	0.40
		Female	50.57% (623)	48.86% (1269)	49.42% (1892)	0.51
		Race/Ethnicity				<0.01
		Non-Hispanic white	52.49% (739)	63.79% (1701)	60.04% (2440)	
		Non-Hispanic Black	18.80% (155)	12.23% (194)	14.41% (349)	
		Hispanic	22.53% (250)	16.87% (341)	18.75% (591)	
		Other	6.18% (108)	7.03% (264)	6.75% (372)	
		Sports activity	52.54% (698)	63.28% (1567)	59.71% (2265)	<0.01
		Hours spent in school-sponsored activities	4.69 (4.70)	4.95 (4.69)	4.86 (4.70)	0.06
		Sedentary lifestyles	17.05 (12.66)	16.07 (13.72)	16.40 (13.37)	0.67
		Perceived weight	3.71 (0.75)	2.82 (0.63)	3.12 (0.79)	<0.01
		Intention to change weight				<0.01
		Lose weight	69.73% (868)	20.26% (510)	36.68% (1378)	
		Gain weight	2.11% (26)	16.29% (407)	11.59% (433)	
		Stay the same weight	12.77% (171)	32.38% (760)	25.87% (931)	
		Not trying to do anything about their weight	12.95% (158)	29.41% (784)	23.96% (942)	
	2. Parenting capacity					
		Parents’ education				<0.01
		Lower than high school	10.40% (122)	5.98% (131)	7.44% (253)	
		High school	24.62% (291)	17.34% (426)	19.76% (717)	
		Greater than high school	64.98% (839)	76.69% (1944)	72.80% (2783)	
		Parents’ employment	5.94 (2.23)	6.09 (2.14)	6.04 (2.17)	0.12
		Parents’ health	3.60 (0.98)	3.84 (0.96)	3.76 (0.97)	<0.01
		Parents’ income	8.27 (3.06)	9.10 (3.07)	8.83 (3.09)	<0.01
		Parent-child relationship	4.92 (1.00)	5.05 (1.01)	5.00 (1.01)	<0.05
		Parental characteristics	58.42% (762)	59.81% (1665)	59.35% (2427)	0.59
	3. Family Function					
		Family structure	72.80% (933)	76.23% (1978)	75.09% (2911)	0.13
		Family size with members less than 18 years old	2.27 (1.10)	2.31 (1.09)	2.30 (1.09)	0.29
		Family TV restriction	84.16% (1055)	85.99% (2108)	85.38% (3163)	<0.05
		Food security	0.77 (2.44)	0.54 (1.88)	0.62 (2.09)	<0.01
		SES quintile	2.77 (1.35)	3.27 (1.39)	3.10 (1.39)	<0.01
	4. School					
		School’s ratio of minorities (degrees)	2.82 (1.53)	2.64 (1.45)	2.70 (1.48)	<0.01
		School reduced-price lunch program	3.18 (.90)	3.08 (.91)	3.11 (.91)	<0.01
		School free lunch program	35.71 (25.00)	30.55 (24.18)	32.27 (24.68)	<0.01
	5. Environment					
		Urbanity				<0.01
		Large city	33.75% (387)	29.33% (704)	30.80% (1091)	
		Mid-size city	36.78% (456)	46.81% (1102)	43.48% (1558)	
		Small town and rural	29.48% (409)	23.86% (695)	25.72% (1104)	

*n* = 3753; weighted *n* = 1,284,209. Source: 2007 Early Childhood Longitudinal Study.

**Table 3 ijerph-18-04779-t003:** PS test comparison between matched and unmatched covariates.

Variables							
Dependent Variables	Overweight	Binary Variable (1 = Overweight, 0 = Normal)	Unmatched (*n* = 2993) Matched (*n* = 351)	Mean		*t*-Test	
				Treated	Control	t	*p* > |t|
Covariates	1. Individual						
		Age	U	13.70	13.69	0.47	0.64
			M	13.71	13.66	1.11	0.27
		Female	U	0.53	0.51	0.90	0.37
			M	0.54	0.53	0.30	0.076
		Race/Ethnicity					
		Non-Hispanic white	U	-	-	-	-
			M	-	-	-	-
		Non-Hispanic Black	U	0.27	0.06	13.53	<0.01
			M	0.24	0.23	0.44	0.66
		Hispanic	U	0.24	0.14	4.90	<0.01
			M	0.25	0.26	−0.17	0.86
		Other	U	0.12	0.08	2.39	<0.05
			M	0.12	0.11	0.36	0.72
		School-sponsored activities	U	4.02	4.99	−3.73	<0.01
			M	4.08	3.79	0.86	0.39
		Sports activity	U	0.60	0.62	−1.00	0.32
			M	0.59	0.54	1.37	0.17
		Sedentary lifestyles	U	20.27	15.88	6.48	<0.01
			M	19.70	20.11	−0.34	0.73
		Perceived weight degrees	U	3.17	3.11	1.42	0.16
			M	3.19	3.11	0.21	0.23
		Intention to change weight					
		1 = lose weight	U	-	-	-	-
			M	-	-	-	-
		2 = gain weight	U	0.14	0.11	1.50	0.13
			M	0.13	0.14	−0.44	0.66
		3 = stay the same weight	U	0.24	0.25	−0.49	0.62
			M	0.24	0.23	0.27	0.79
		4 = not trying to do anything about	U	0.21	0.26	2.27	<0.05
		their weight	M	0.21	0.26	−1.51	0.13
	2. Parenting capacity						
		Parents’ education					
		Lower than high school	U	-	-	-	-
			M	-	-	-	-
		High school	U	0.36	0.17	8.89	<0.01
			M	0.35	0.34	0.16	0.87
		Greater than high school	U	0.39	0.73	−13.69	<0.01
			M	0.40	0.38	0.70	0.49
		Parents’ employment	U	5.34	6.33	−9.37	<0.01
			M	5.38	5.29	0.53	0.60
		Parents’ health	U	3.38	3.82	−8.39	<0.01
			M	3.38	3.40	−0.22	0.83
		Parents’ income	U	6.12	9.34	−21.13	<0.01
			M	6.27	6.20	0.34	0.73
		Parent-child relationship	U	4.73	5.13	−7.83	<0.01
			M	4.74	4.62	1.61	0.11
		Parental characteristics	U	0.41	0.69	−10.62	<0.01
			M	0.42	0.44	−0.30	0.76
	3. Family Function						
		Family structure	U	0.60	0.81	−9.53	<0.01
			M	0.60	0.60	0.15	0.88
		Family size with members less than 18	U	2.50	2.25	4.38	<0.01
		years old	M	2.49	2.52	−0.31	0.76
		Family TV restriction	U	0.82	0.86	−2.03	<0.05
			M	0.82	0.84	−0.81	0.42
		Food security	U	1.41	0.50	8.55	<0.01
			M	1.36	1.29	0.30	0.77
		SES	U	2.00	3.31	18.09	<0.01
			M	2.05	2.01	0.41	0.69
	4. School						
		School’s ratio of minorities	U	3.26	2.56	9.13	<0.01
			M	3.21	3.10	0.87	0.39
		School reduced-price lunch program	U	3.24	3.08	3.27	<0.01
			M	3.27	3.33	−0.96	0.34
		School free lunch program	U	45.65	29.09	14.10	<0.01
			M	45.48	44.45	0.54	0.59
	5. Environment						
		Urbanity					
		Large city	U	-	-	-	-
			M	-	-	-	-
		Mid-size city	U	0.25	0.44	−7.25	<0.01
			M	0.26	0.24	0.61	0.54
		Small town and rural	U	0.46	0.28	7.01	<0.01
			M	0.43	0.44	−0.30	0.76

Source: 2007 Early Childhood Longitudinal Study.

**Table 4 ijerph-18-04779-t004:** Weighted logistic regression after PSM.

Variables				
Dependent Variables	Overweight	Binary variable (1 = overweight, 0 = normal)	OR	CI
Treatment	Head Start	Binary variable	1.28	0.73–2.27
Covariates	1. Individual			
		Age	0.91	0.55–1.49
		Female	0.86	0.50–1.48
		Race/Ethnicity		
		Non-Hispanic white	-	-
		Non-Hispanic Black	5.23 **	2.04–13.37
		Hispanic	1.32	0.51–3.42
		Other	1.28	0.29–5.61
		School-sponsored activities	1.07 *	1.01–1.15
		Sports activities	0.32 **	0.17–0.61
		Sedentary lifestyles	1.00	0.99–1.01
		Perceived weight degrees	6.86 **	4.22–11.16
		Intention to change weight		
		1 = lose weight	-	-
		2 = gain weight	0.04 **	0.01–0.15
		3 = stay the same weight	0.08 **	0.04–0.16
		4 = not trying to do anything about their weight	0.15 **	0.08–0.30
	2. Parenting capacity			
		Parents’ education		
		Lower than high school	-	-
		High School	1.76	0.87–3.54
		Greater than high school	1.19	0.56–2.50
		Parent’s employment	0.99	0.81–1.21
		Parents’ health	1.07	0.79–1.45
		Parents’ income	1.05	0.91–1.22
		Parent-child relationship	1.02	0.78–1.34
		Parental characteristics	0.87	0.39–1.92
	3. Family Function			
		Family structure	1.46	0.52–4.08
		Family size with members less than 18 years old	0.78 *	0.63–0.97
		Family TV restriction	0.40 *	0.20–0.80
		Food security	0.97	0.85–1.10
		SES	0.87	0.59–1.29
	4. School			
		School’s ratio of minorities	1.15	0.84–1.56
		School free lunch program	1.01	1.01–1.02
		School reduced-price lunch program	1.13	0.82–1.57
	5. Environment			
		Urbanity		
		Large city	-	-
		Mid-size city	1.76	0.82–3.78
		Small town and rural	1.54	0.71–3.33
Total	*n*		702	
	Weighted *n*		274,253	

* *p* < 0.05; ** *p* < 0.01; Source: 2007 Early Childhood Longitudinal Study.

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
