# Peer review of "Effects of the Head Start Program on Socioecological Obesogenic Factors in American Children"

_ijerph, 2021, doi:10.3390/ijerph18094779_

Round 1

Reviewer 1 Report

2.3 billion children and adults are considered to be overweight globally, with obesity costs exceeding $700 billion each year. This article reveals that multiple dimensions of sociological obesogenic factors including individual, parental, familial, and community support factors affect the weight of children from low-income families and should be considered when establishing behavioral and policy interventions to thwart the child- hood obesity epidemic. These results are of great theoretical significance and practical value for solving overweight and obesity among children. The analysis process is comprehensive, good organized, large amount of information and so on. Minor revision can be published in International Journal of Environmental Research and Public Health. However, there are some major issues need to be improved:

  1. Abstract:Emphasis should be placed on the importance of diet foods;
  2. Introduction: Updated reference to supplement refined staple foods leading to obesity,such as https://www.hindawi.com/journals/omcl/2020/3836172/
  3. Materials and Methods: The references section can be refined;
  4. Results: In Head Start Program, The most important factors affecting obesity should be identified;
  5. Discussion: The role of diet staples such as barley functional foods in obesity and overweight interventions should be increased;
  6. Conclusions: Increased diet in future studies to combat obesity;There are usually no references in the conclusions;
  7. References: Authors shouldrevise the format of reference according to IJERPH Journal.

Author Response

Response to Reviewer 1 Comments

We really appreciate the great comments and suggestions for the better research.

Point 1. Abstract: Emphasis should be placed on the importance of diet foods

Response 1:  In this study, the nutritional factor for obesity was also confirmed in the head start program, but it was not statistically significant. The effect of diet on obesity is very self-evident, but this study did not emphasize the importance of diet.

Point 2. Introduction: Updated reference to supplement refined staple foods leading to obesity, such as https://www.hindawi.com/journals/omcl/2020/3836172/

Response 2: Thank you so much for your reference. The effects of obesity on food have been confirmed in many studies. Therefore, the most cited papers were used as the references used in this paper.

Point 3. Materials and Methods: The references section can be refined.

Response 3: The data used in this study are part of a longitudinal study conducted every 10 years. So it takes a lot of time to get a complete 10 years worth of data. The data used in this study was collected more than 10 years ago, but it can be used as public data within a few years. Therefore, the materials and research methods for the data used in this study are up-to-date.

Point 4. Results: In Head Start Program, the most important factors affecting obesity should be identified.

Response 4: The impact of the head start program on child obesity was included in the paper line 164 to 166 results report.

Point 5. Discussion: The role of diet staples such as barley functional foods in obesity and overweight interventions should be increased.

Response 5: Thank you for your valuable comment. Many papers and studies have confirmed that diet has an effect on obesity and emphasized the need for intervention programs. Therefore, in this study, it was omitted because it was not considered the core part of the study.

Point 6. Conclusions: Increased diet in future studies to combat obesity; There are usually no references in the conclusions.

Response 6: Thank you for your valuable comment. Unless it is a special case, references are not mentioned in the conclusion of the study, but this study is to verify the effectiveness of the child welfare and well-being program called Head Start.

Point 7. References: Authors should revise the format of reference according to IJERPH Journal.

Response 7: As you mentioned, lines 308 through 312 of the bibliography paper have been edited in red to match the format of the journal.

Reviewer 2 Report

Effects of the Head Start Program on Socioecological Obesogenic Factors in American Children

Int. J. of Environmental Research and Public Health

“The purpose of this study was to determine if the Head Start program affects children’s BMI status with respect to socio ecological obesogenic factors through PSM analysis.”

Thank you for the opportunity to review this manuscript. The authors present a thought provoking manuscript on a useful and timely topic. It is clearly written however, this reviewer is left with several questions and suggestions which follow.

Introduction: Many of the references appear to be older (2016 or earlier) and may not reflect current information, assuming other work in this area has been published. For example, reference 5, Flegal et al 2010 is cited on line 35 regarding 1/3 of adults in the US are overweight and/or obese. In a quick Google search I found “Obesity in America: Overweight and Obesity Statistics in 2021” that states, “Nearly 40% of American adults aged 20 and over are obese. 71.6% of adults aged 20 and over are overweight, including obesity. (National Health and Nutrition Examination Survey, 2017-2018; Harvard School of Public Health, 2020).”  Similarly information on 2017-2018 rates among children is available.

Materials and Methods:  How were variables selected to evaluate “quality of life”?

How were subjects “recruited” from the ECLS-K data sets?

How old were the children when these factors were assessed in this cross sectional study?

At what age was obesity evaluated- during prekindergarten or at 8th grade?  If the same child was measured more than once, which measure was used here?

Several values are given for the number of participants – 3753, 2993 (unmatched), 702 (Table 4), 448, and 351. Would a Table listing the number of subjects at each phase of analysis be useful?

What was the age range, if 13.69 is the mean? On page 3 the authors state, “Participants range in age from 2 to 13”.  How do the authors account for 2 year olds playing computer or video games? Please clarify the age of the children when the assessments were done.

Define PSM the first time it is used and describe the method briefly in more detail.

Page 2 paragraph 3, line 7 please spell out FPL here or where it is first used.

Page 3 para 2, line 2 (e.g…..) is confusing. How is normal weight defined? Do you wish to differentiate between overweight, including obesity, or only obesity?  Did you exclude underweight children?

Ibid, line 1 “the main binary variable was overweight (e.g., equal or greater than 95th percentile). This would be the definition of obesity. The term overweight is used in Tables 2 and 3. In Table 4, Overweight is now equal or greater than 85th percentile.  Why was the definition changed and has the change altered the outcomes in this manuscript? Please clarify.

Page 4, para 1, lines 5-6 both 0 and 1 are defined as indicating parental care. Please clarify.

Ibid, line 8 please clarify “The number of families under the age of 18”. Is this number of families with children under the age of 18?

Page 8 para 2, line 2 Please define use of “nest”.  It is a term I haven’t seen in this context but perhaps it is used in schools.

Ibid, para 4, line 6 please clarify that the 448 of the sample (not of the girls) participated in Head Start.

Page 6 para 1 Could you please explain why only 351 participants were matched and not 448? Further, how are the nonparticipants selected from the 3,000+ remaining pool?

Pages 6&8, Subheadings 3.1 and 3.2 are identical. Please differentiate.

Discussion: The authors acknowledge limitations appropriately.

Re small sample size, the number of Non-Hispanic Blacks is lowest (N=349). Would rephrasing (e.g., Hispanic and others) to (e.g., non-white groups) be appropriate?

 In offering unobserved confounding socio ecological factors related to childhood obesity, did the authors consider assessing or listing for future exploration  “Adverse Childhood Experiences”? Does the ECLS-K data set address that?

General comments:

Does Table 1 add insight into this manuscript?

The lack of line numbering after line 44 has made targeted comments more challenging.

I recommend review of the manuscript by a statistician.

Author Response

Response to Reviewer 1 Comments

We really appreciate the great comments and suggestions for the better research.

Point 1: Introduction: Many of the references appear to be older (2016 or earlier) and may not reflect current information, assuming other work in this area has been published. For example, reference 5, Flegal et al 2010 is cited on line 35 regarding 1/3 of adults in the US are overweight and/or obese. In a quick Google search I found “Obesity in America: Overweight and Obesity Statistics in 2021” that states, “Nearly 40% of American adults aged 20 and over are obese. 71.6% of adults aged 20 and over are overweight, including obesity. (National Health and Nutrition Examination Survey, 2017-2018; Harvard School of Public Health, 2020).”  Similarly information on 2017-2018 rates among children is available.

Response 1:

  •  As your dearest words, we've changed the obesity rate of American adults from lines 315 to 316 with the latest reference.
  • Ogden CL, Fryar CD, Martin CB, et al. Trends in obesity prevalence by race and hispanic origin - 1999-2000 to 2017-2018. JAMA - J Am Med Assoc. 2020;324(12):1208-1210.

Materials and Methods: 

Point 2: How were variables selected to evaluate “quality of life”?

Response 2:

  • Thank you for your valuable comment. Studies related to the quality of life of children are representatively referring to depression mentally and obesity physically. Therefore, this study used the child's obesity as a variable related to the children's quality of life.
  • (Bhadoria A, Sahoo K, Sahoo B, Choudhury A, Sufi N, Kumar R. Childhood obesity: Causes and consequences. J Fam Med Prim Care. 2015;4(2):187. doi:10.4103/2249-4863.154628
  • Russell-Mayhew S, McVey G, Bardick A, Ireland A. Mental health, wellness, and childhood overweight/obesity. J Obes. 2012;2012:281801. doi:10.1155/2012/281801
  • Schwimmer JB, Burwinkle TM, Varni JW. Health-Related Quality of Life of Severely Obese Children and Adolescents. J Am Med Assoc. 2003;289(14):1813-1819. doi:10.1001/jama.289.14.1813)

Point 3: How were subjects “recruited” from the ECLS-K data sets?

Response 3: Thank you for your valuable point.The selection of study participants is detailed in lines 73 to 77 of the paper with red.

Point 4: How old were the children when these factors were assessed in this cross sectional study?

Response 4: The data for this study are from the last data set from the longitudinal study. Therefore, the children used in the study were children who participated in the Head Start program when they were young, and the age at the time was between 12 and 14 years of age.

Point 5: At what age was obesity evaluated-during prekindergarten or at 8th grade?  If the same child was measured more than once, which measure was used here?

Response 5: Binary factors were used to determine whether 8th graders were obese.

Point 6: Several values are given for the number of participants – 3753, 2993 (unmatched), 702 (Table 4), 448, and 351. Would a Table listing the number of subjects at each phase of analysis be useful?

Response 6: A total of 702 children were analyzed with 351 children related to participation in the head start, and the number of participants was 274,253 due to the statistics determined by weighting this by the total population, and it is not useful to describe the number of people at each stage.

Point 7: What was the age range, if 13.69 is the mean? On page 3 the authors state, “Participants range in age from 2 to 13”.  How do the authors account for 2 year olds playing computer or video games? Please clarify the age of the children when the assessments were done.

Response 7: Thanks for the good point. The 8th grade is officially 13 years old. However, as we organized this study by grade, some children were 15 years old. So, as you said, I corrected my age to 15 years old.

Point 8: Define PSM the first time it is used and describe the method briefly in more detail.

Response 8: Thank you for your valuable comment. We put the definition of PSM in red on paper line 155.

Point 9: Page 2 paragraph 3, line 7 please spell out FPL here or where it is first used.

Response 9: We spelled out FPL in red on paper line 75.

Point 10: Page 3 para 2, line 2 (e.g…..) is confusing. How is normal weight defined? Do you wish to differentiate between overweight, including obesity, or only obesity?  Did you exclude underweight children?

Response 10: Thank you for your valuable comments. The binary factor of overweight used in this study is that overweight includes overweight and obesity, and normal weight does not include underweight. None of the 701 participants in this study were underweight.

Point 11: Ibid, line 1 “the main binary variable was overweight (e.g., equal or greater than 95th percentile). This would be the definition of obesity. The term overweight is used in Tables 2 and 3. In Table 4, Overweight is now equal or greater than 85th percentile.  Why was the definition changed and has the change altered the outcomes in this manuscript? Please clarify.

Response 11: You've edited it like you did. Normal weight is 5% or more and 84% or less, and overweight is 85% or more in the line of 94 to 95 on the paper. We checked the statistical analysis again for a double check. It's just a typo. Thank you.

Point 12: Page 4, para 1, lines 5-6 both 0 and 1 are defined as indicating parental care. Please clarify.

Response 12: It is a family form of parents that are mother and father parents and not. In other words, they are single mothers, single parents, or children who are adopted and raised.

Point 13: Ibid, line 8 please clarify “The number of families under the age of 18”. Is this number of families with children under the age of 18?

Response 13: Yes, this refers to the number of family members who live together under the age of 18.

Point 14: Page 8 para 2, line 2 Please define use of “nest”.  It is a term I haven’t seen in this context but perhaps it is used in schools.

Response 14: Sorry, but I couldn't find 'nest' in the second peragraph on page 8 you pointed out. However, as you said in the variable description used, it means that the children belong to school. Thank you.

Point 15: Ibid, para 4, line 6 please clarify that the 448 of the sample (not of the girls) participated in Head Start.

Response 15: 448 came out of a total of 702 children. Next, the effectiveness of 702 people was verified by participating in the head start by 351 people each.

Point 16: Page 6 para 1 Could you please explain why only 351 participants were matched and not 448? Further, how are the nonparticipants selected from the 3,000+ remaining pool?

Response 16: When analyzing the characteristics of the participant children, it was not considered whether they participated in the head start program. The rest of the analysis is analyzed by the statistics program automatically divided into 1:1 head start gram for each 351 people.

Point 17: Pages 6&8, Subheadings 3.1 and 3.2 are identical. Please differentiate.

Response 17: We made it different in red. Thank you.

Discussion: 

Point 18: Re small sample size, the number of Non-Hispanic Blacks is lowest (N=349). Would rephrasing (e.g., Hispanic and others) to (e.g., non-white groups) be appropriate?

Response 18: Following your valuable comments, we edited paper line 276 in red.

Point 19: In offering unobserved confounding socio ecological factors related to childhood obesity, did the authors consider assessing or listing for future exploration “Adverse Childhood Experiences”? Does the ECLS-K data set address that?

Response 19: The ECLS-K program provides national representative data on children’s status at birth and at various points, including their transitions to nonparental care, on early education programs and schools, and on children’s experiences and growth through the eighth grade. The ECLS-K data were collected from children, their parents, teachers, and schools through a variety of methods such as one-on-one assessments, computer-assisted telephone interviews (CATI), and self-reported paper-and-pencil questionnaires. The ECLS-K program also provides data regarding the relationships among a wide range of family, school, community, and individual variables with regard to the children’s development, early learning, and performance in school. It is questionable whether the ECLS-K data set will be able to measure children's overall grievance experiences.

Reviewer 3 Report

The work deals with an interesting topic that is current in many countries.The topic of the study is presented very clearly. Overall it is a well written and well exposed work.There are some points to review.

Point 1: The data collected are too old, they were collected in 2007! Since so much time has elapsed between collection and analysis, can this be a limitation of the work? Could some parameters have changed in 14 years? Why was such old data used?
Point 2: In some cases (lines 127-139 and 144-152) there is text in red.
Point 3: In the paragraph of the statistical analysis (2.5) it would be appropriate to insert other references, perhaps from other works that have used the PSM analysis, in order to expand the paragraph. The same is true for Logistic Regression.
Point 4: References is not very updated, it is advisable to insert more recent works and as suggested before, some references relating to the methodology used, as I have not seen any references about it.

Author Response

Response to Reviewer 3 Comments

We really appreciate the great comments and suggestions for the better research.

Point 1: The data collected are too old, they were collected in 2007! Since so much time has elapsed between collection and analysis, can this be a limitation of the work? Could some parameters have changed in 14 years? Why was such old data used?

Response 1: The data used in this study are part of a longitudinal study conducted every 10 years. So it takes a lot of time to get a complete 10 years’ worth of data. The data used in this study was collected more than 10 years ago, but it can be used as public data within a few years. Therefore, the materials and research methods for the data used in this study are up-to-date.

Point 2: In some cases (lines 127-139 and 144-152) there is text in red.

Response 2: Thank you for valuable comments, it has been modified at the request of other reviewers.

Point 3: In the paragraph of the statistical analysis (2.5) it would be appropriate to insert other references, perhaps from other works that have used the PSM analysis, in order to expand the paragraph. The same is true for Logistic Regression.

Response 3:  As you said, we put the references of those statistical method in red.

Point 4: References is not very updated, it is advisable to insert more recent works and as suggested before, some references relating to the methodology used, as I have not seen any references about it.

Response 4: We have updated some references in red. Thank you so much.